# The Utilization of Machine Learning Algorithms for Assisting Physicians in the Diagnosis of Diabetes

**DOI:** 10.3390/diagnostics13122087

**Published:** 2023-06-16

**Authors:** Linh Phuong Nguyen, Do Dinh Tung, Duong Thanh Nguyen, Hong Nhung Le, Toan Quoc Tran, Ta Van Binh, Dung Thuy Nguyen Pham

**Affiliations:** 1School of Preventive Medicine and Public Health, Ha Noi Medical University, 1, Ton That Tung Street, Dong Da District, Ha Noi 100000, Vietnam; nplinh239@gmail.com; 2Vietnam Diabetes Educators Association, 52/A1 Dai Kim Urban Area, Hoang Mai District, Ha Noi 100000, Vietnam; binhnoitiet@gmail.com; 3Saint Paul General Hospital, 12A Chu Van An, Ba Dinh District, Ha Noi 100000, Vietnam; 4Institute for Tropical Technology, Vietnam Academy of Science and Technology (VAST), 18 Hoang Quoc Viet St., Cau Giay Dist., Ha Noi 100000, Vietnam; ntduong182@gmail.com; 5Institute of Natural Products Chemistry, Vietnam Academy of Science and Technology (VAST), 18 Hoang Quoc Viet St., Cau Giay Dist., Ha Noi 100000, Vietnamtranquoctoan2010@gmail.com (T.Q.T.); 6NTT Institute of Applied Technology and Sustainable Development, Nguyen Tat Thanh University, Ho Chi Minh City 70000, Vietnam; 7Faculty of Environmental and Food Engineering, Nguyen Tat Thanh University, Ho Chi Minh City 70000, Vietnam

**Keywords:** diabetes, detection, diabetes prediction, machine learning

## Abstract

This paper investigates the use of machine learning algorithms to aid medical professionals in the detection and risk assessment of diabetes. The research employed a dataset gathered from individuals with type 2 diabetes in Ninh Binh, Vietnam. A variety of classification algorithms, including Decision Tree Classifier, Logistic Regression, SVC, Ada Boost Classifier, Gradient Boosting Classifier, Random Forest Classifier, and K Neighbors Classifier, were utilized to identify the most suitable algorithm for the dataset. The results of the present study indicate that the Random Forest Classifier algorithm yielded the most promising results, exhibiting a cross-validation score of 0.998 and an accuracy rate of 100%. To further evaluate the effectiveness of the selected model, it was subjected to a testing phase involving a new dataset comprising 67 patients that had not been previously seen. The performance of the algorithm on this dataset resulted in an accuracy rate of 94%, especially the study’s notable finding is the algorithm’s accurate prediction of the probability of patients developing diabetes, as indicated by the class 1 (diabetes) probabilities. This innovative approach offers a meticulous and quantifiable method for diabetes detection and risk evaluation, showcasing the potential of machine learning algorithms in assisting clinicians with diagnosis and management. By communicating the diabetes score and probability estimates to patients, the comprehension of their disease status can be enhanced. This information empowers patients to make informed decisions and motivates them to adopt healthier lifestyle habits, ultimately playing a crucial role in impeding disease progression. The study underscores the significance of leveraging machine learning in healthcare to optimize patient care and improve long-term health outcomes.

## 1. Introduction

Type 2 diabetes mellitus (T2DM) is a chronic metabolic disorder characterized by elevated levels of blood glucose due to inadequate insulin secretion or impaired insulin action. The International Diabetes Federation (IDF) estimates that 536.6 million (10.5%) adults globally will be diagnosed with T2DM in 2021, and the number will grow to 783.2 million (12.2%) by 2045 [1]. In Vietnam, diabetes is becoming increasingly common and is now projected to affect one in every twenty Vietnamese individuals [2]. Diabetes is one of the leading causes of death and has been associated with an increased risk of various complications, including cardiovascular diseases, stroke, renal failure, blindness, and neuropathy. The disease imposes a significant economic burden on healthcare systems, and the cost of treating diabetes-related complications further exacerbates this burden. Diabetes-related healthcare expenditure was estimated to be $966 billion worldwide and $1.7 billion in Vietnam in 2021 [3,4]. These burdens of diabetes underscore the importance of early detection and management to minimize disease progression, reduce the risk of costly complications, and improve patient outcomes. In this regard, artificial intelligence (AI) and machine learning (ML) have emerged as promising approaches for early diabetes detection.

AI entails the development of algorithms and computer systems that mimic human intelligence, enabling them to learn from data and experiences and perform intricate tasks that typically require human intelligence [5]. ML, a subfield of AI, focuses on developing algorithms that learn from data without explicit programming, particularly in decision-making tasks. ML employs statistical techniques to uncover patterns from large and complex datasets, facilitating accurate predictions and automated decision-making. ML’s ability to handle intricate relationships and nonlinear patterns distinguishes it from traditional statistical methods, making it a valuable tool in the healthcare domain [6]. The utilization of algorithms and formulas for monitoring processes and applications finds application across diverse fields, including physics and chemistry [7,8,9,10]. These approaches are employed for purposes such as early diagnosis, predictive analysis, and prognostication. The applications of AI and ML in early disease detection and personalized health monitoring are extensive, with numerous studies showcasing their potential to improve disease detection and diagnosis across various conditions, such as cancer, cardiovascular disease, and Alzheimer’s disease [11,12,13,14].

Specifically, regarding early diabetes detection, ML has emerged as a promising approach for the detection of diabetes and its complications. One specific example is the study conducted by Shukla (2020) in which a Logistic Regression algorithm was used to predict the risk of developing diabetes in Indian adults using demographic and clinical variables, such as glucose, body mass index (BMI), and pregnancies. The study reported a high accuracy rate of 82.92%, suggesting that the developed model could identify individuals at risk of developing diabetes and potentially prevent its onset [15]. Another example is the study by Islam et al. (2020) that employed serval ML algorithms, including Naive Bayes, Logistic Regression, and Random Forest algorithms, to predict the risk of diabetes in a sample of 520 individuals. The study found that the Random Forest algorithm provided the best result with an accuracy of 99% [16]. In addition, Kavakiotis et al. (2017) developed ML algorithms to predict the development of type 2 diabetes in patients with impaired glucose tolerance using data from 11,000 patients. The algorithms achieved an accuracy of 90%, which outperformed traditional statistical methods [17]. These studies demonstrated the potential of ML to improve the accuracy of diabetes risk prediction and facilitate earlier interventions, ultimately leading to improved patient outcomes.

However, in the context of diabetes detection in Vietnam, there is currently a paucity of studies exploring the application of artificial intelligence (AI) and machine learning (ML) techniques. Despite the increasing adoption of AI and ML methods in healthcare, particularly in diabetes detection, worldwide, the specific application of these approaches within the Vietnamese population remains largely unexplored. The absence of studies investigating AI and ML for diabetes detection in Vietnam remains a significant research gap. Therefore, our research aims to investigate the application of ML for detecting diabetes in Vietnam using data specifically sourced from Vietnamese patients. We employed various classification algorithms, such as Decision Tree Classifier, Linear Regression, Support Vector Classifier (SVC), Adaboost Classifier, Gradient Boosting Classifier, Random Forest, and K Nearest Neighbor, to predict diabetes in patients. We evaluated the performance of these classification methods using different metrics and selected the best-fitting model for predicting new, unseen data, which was then compared with the doctor’s diagnosis. Our study also explores the use of probabilities of the results as quantitative measures for diabetes diagnosis and risk assessment, which can provide practical and actionable information to healthcare professionals, enhancing their clinical utility.

Overall, our research aims to contribute to the development of ML-based approaches that can be effectively utilized as an assistance tool for medical experts in the diagnosis and management of diabetes in Vietnamese patients, enabling clinicians to help patients better understand their condition, offer personalized interventions and treatments based on individual risk profiles, which would improve patient outcomes and quality of life.

## 2. Materials and Methods

The objective of this study is to develop a machine-learning model for early-stage diabetes prediction by utilizing a dataset collected exclusively from Vietnamese patients. The following section outlines the implementation and procedures involved in designing the proposed diabetes prediction system. Figure 1 illustrates the model diagram of the proposed system.

### 2.1. Model Selection

#### 2.1.1. Data Collection

In this study, data were collected from the Lifestyle intervention trial program, which aimed to prevent type 2 diabetes in the Northern province of Ninh Binh, Vietnam. The participants in the study were individuals aged 18 or older of both genders who had a high risk of developing diabetes based on their responses to a questionnaire assessing their risk score. Standardized tools and techniques were used to measure the participant’s physical characteristics and biochemical measurements.

The dataset included 2153 patients, each with 14 attributes, such as Age, Gender, BMI, Insulin level, Diastolic Blood pressure, Systolic Blood pressure, Fasting plasma glucose, Plasma glucose after 2 h, Total cholesterol, Triglycerides, High-Density Lipoprotein (HDL) Cholesterol, Waist circumference, Hip circumference, and Outcome (Table 1). The target variable of the study was the ‘Outcome’ attribute, which is binary, with 0 indicating non-diabetes and 1 indicating diabetes. The remaining 13 attributes were considered independent variables.

In this study, the analysis of the diabetes dataset was performed using Scikit-learn, a widely-used open-source Python library for machine learning. Scikit-learn offers a variety of tools for implementing machine learning algorithms like classification, regression, and clustering, as well as tools for data preprocessing, model selection, and evaluation. By leveraging Scikit-learn, we were able to efficiently and effectively train and evaluate multiple machine-learning models using a uniform interface. Python, with its intuitive syntax, was used to code with Scikit-learn, making it easy to implement complex machine-learning workflows [18].

#### 2.1.2. Data Preprocessing

1.Missing value identification

The dataset contained some missing values and outliers, which could potentially affect the accuracy of our analysis. This was due to dropouts and not being able to conduct blood tests due to resource restraints. To address this issue, we decided to remove these problematic samples from our dataset. By doing so, we aimed to minimize any potential bias or errors introduced during data processing and ensure that our analysis was based on reliable and complete data.

2.Key features identification

Table 2 and Figure 2 present the results of using Pearson’s correlation coefficient to determine the significance of each attribute in the dataset. Pearson’s correlation coefficient is a widely used method for identifying the most influential attributes or features. This approach involves calculating a correlation coefficient that quantifies the relationship between the input and output attributes in the dataset, with a coefficient value ranging from −1 to 1. A coefficient value above 0.5 or below −0.5 indicates a significant correlation, while a value of zero indicates no correlation [19].

Identifying the important features is crucial in enhancing the effectiveness of the model. The relationship between the input and output variables is important in the context of diabetes diagnosis using artificial intelligence, as it helps to comprehend the inter-dependency among the variables and their influence on the outcome, i.e., diabetes. The correlation matrix can assist in recognizing the significant input variables that have a substantial correlation with the output variable and may be beneficial in predicting diabetes. This can enhance the efficiency and effectiveness of artificial intelligence in diabetes diagnosis.

Observations show that characteristics, such as 2-h plasma glucose, Fasting plasma glucose, and Waist circumference, are most closely associated with the outcome (Table 3). This finding is consistent with previous research; especially, it is noteworthy that waist circumference has been shown to be a potentially important predictor of diabetes risk, particularly in Asian populations [20,21,22].

3.Oversampling

Upon completion of the preprocessing stage, we were left with a dataset consisting of 500 instances, out of which 449 corresponded to non-diabetic cases, while the remaining 51 belonged to diabetic cases. The imbalanced distribution of the two classes may cause difficulties for the models in detecting diabetic cases accurately. In such situations, the models tend to predict the majority class more frequently, which in our case is class 0 (non-diabetes), leading to high overall accuracy but poor performance in detecting class 1 (diabetes).

To tackle the issue of imbalanced classes in the diabetes dataset, we adopted Random Over-Sampling (ROS) technique. The method entails duplicating instances randomly from the minority class (class 1—diabetes) so that it is balanced with the majority class (class 0—non-diabetes). The goal is to create a more balanced dataset that allows machine learning models to learn equally from both classes, hence improving the model’s accuracy in detecting class 1.

4.Data splitting

Once the data had been preprocessed and cleaned, the dataset was deemed suitable for training and testing. To achieve this, we divided the dataset into two parts: the training set, which constituted 70% of the data, was used to train the model, allowing it to learn patterns and relationships, while the test set, which accounted for 30% of the data, served as an independent dataset to evaluate the model’s performance and generalization ability. The 70–30 split, widely employed in machine learning, strikes a balance between an adequate training set size and a sufficiently large test set, ensuring a robust evaluation of the model’s performance [23,24].

After splitting, the size of the training dataset was 8164 data for 628 people with 449 cases of diabetes and 449 non-diabetes. The size of the testing dataset was 3510 data for 270 people with 133 cases of diabetes and 137 non-diabetes.

5.Selection of classification models

In this study, we compared the performance of seven different machine learning classification algorithms, namely Decision Tree Classifier [25,26], Logistic Regression [27], Support Vector Classification (SVC) [28], Ada Boost Classifier [29,30], Gradient Boosting Classifier [31], Random Forest Classifier [25], and K-Neighbors Classifier [6,23,32] algorithm. The purpose of this analysis was to determine which algorithm is most appropriate for our dataset and to generate predictions for new data. We evaluated the performance of each algorithm using various metrics.

6.Validation of classification models

The K-fold cross-validation method and the accuracy metric were employed to evaluate each algorithm’s performance.

The method of K-fold cross-validation is widely used to assess machine learning models. The data set is divided into k equal sections, where k is a user-defined number. The model is then evaluated on the remaining portion after training on k−1 of these parts. Each part serves as the test set once throughout this procedure’s repetition of k times. An overall assessment of the model’s performance is provided by averaging the performance measures collected throughout each iteration [33].

In our study, we selected k = 10, which resulted in the data being split into ten subsets. The model was trained on nine of these subsets and evaluated on the remaining subset, with this process being repeated ten times using a different subset as the test set each time. The resulting performance metrics were then averaged to provide an estimate of the model’s performance.

The accuracy metric was computed by using Scikit-learn’s classification report function, which calculated the number of correctly classified instances and divided it by the total number of instances in the dataset [34].

7.Hyperparameter tuning

After analyzing the performance of our machine learning models, we went on to improve the hyperparameters of the best-performing model. Tuning hyperparameters is an important stage in the development of machine learning models. Hyperparameters are settings that may be changed to increase the performance of an algorithm. We can considerably increase the model’s accuracy and overall performance by determining the ideal combination of hyperparameters [35].

The hyperparameters of each algorithm were fine-tuned in this study using grid search, which is a technique that performs an exhaustive search over a pre-defined range of hyperparameters to identify the combination that results in the best performance for the algorithm [35]. To fine-tune the hyperparameters of each machine learning model, we employed a grid search method where we defined a range of hyperparameters to explore and fitted the model with each combination of hyperparameters. The model was fitted on the training set, and its performance was evaluated on the validation set. This process was repeated for each hyperparameter combination of the model.

To evaluate the performance of the model with different hyperparameters, we used the Stratified K-Fold cross-validation technique. This method involves dividing the dataset into five equal parts or folds. We trained the models on 4 of these folds and tested them on the remaining fold. This process was repeated 5 times, with each fold serving as the test set exactly once. We selected the hyperparameters that produced the best performance based on the highest accuracy.

### 2.2. Machine Learning Model’s Performance in Assisting Diabetes Diagnosis

After tuning the model, we tested its performance on a new dataset containing similar features as the previous dataset. The new dataset consisted of data from 67 patients, where 34 patients had diabetes, and 33 patients did not have diabetes. Prior to testing the model, we preprocessed the new dataset using the same methods as the previous dataset, such as one-hot encoding categorical features, imputing missing values, and scaling the data using the StandardScaler from Scikit-learn. This was performed to ensure that the new dataset had similar formatting and feature scaling as the previous dataset, which allowed for consistent testing of the model.

After preprocessing the new dataset, the optimized model was used to predict the diabetes status of each patient. The model was loaded using the Pandas library, and the Outcome column was removed. The prediction method was applied to obtain the model’s predictions for each patient in the new dataset. The true diabetes status of each patient was compared with the model’s predictions to evaluate the model’s performance. Metrics such as accuracy, precision, recall, and f1 score were used to measure the performance of the optimized algorithm.

Accuracy, as aforementioned, is a performance metric used in the evaluation of a classification model that measures the proportion of correct predictions made by the model. It is computed by dividing the number of correct predictions by the total number of predictions made by the model [34,36]. Precision is a metric that quantifies the proportion of positive predictions that are actually positive, measuring the classifier’s capacity to avoid making incorrect positive predictions. When the precision score is high, it means that the classifier has a low false positive rate. The precision score is obtained by dividing the number of true positives by the sum of true positives and false positives [36]. The recall is a metric used in classification that measures the ability of a model to identify all positive instances in a dataset. It is calculated by dividing the number of true positives by the sum of true positives and false negatives. In other words, it is the proportion of actual positive instances that are correctly predicted by the model. A high recall score indicates that the classifier has a low rate of false negatives [36]. The f1-score is a score that combines precision and recalls into a single metric by calculating their harmonic mean. This score provides a balanced evaluation of the classifier’s performance in terms of both precision and recall [36]. The support in a classification report refers to the number of instances in each class in the original dataset. It is included in the report to provide information on the distribution of classes in the data [27]. The report also provides an average score for each metric, which is weighted by the number of samples in each class. This weighted average is particularly helpful when the classes are not evenly distributed.

The following equations are employed to determine the performance of the classification method.
$$Accuracy=\frac{\left(TP+TN\right)\;}{\left(TP+TN+FP+FN\right)}$$
$$Precision=\frac{TP}{\left(TP+FP\right)}$$
$$Recall=\frac{TP}{\left(TP+FN\right)}$$
$$f1-score=2\times \left(\frac{\left(precision\times recall\right)\;}{\left(precision+recall\right)}\right)$$
where:

*TP* represents the true positives or the cases where the model predicted a positive class and the actual class was also positive.

*FP* are the false positives or the cases where the model predicted a positive class, but the actual class was negative.

*TN* are the true negatives or the cases where the model predicted a negative class and the actual class was also negative.

*FN* represents the false negatives are the cases where the model predicted a negative class, but the actual class was positive.

### 2.3. Quantification of Diabetes Risk and Application in Assisting Diabetes Diagnosis

In this study, the accuracy of the probability estimates produced by the tuned model was tested for predicting the probability of patients developing diabetes. The model was tested with data from four new patients, and the predict_proba method was used to obtain the probability estimates for each patient. In binary classification models like the one used in this study, each class is assigned a probability score between 0 and 1. For patients classified as non-diabetic (class 0), the probability of belonging to class 0 is always greater than 0.5. However, if the probability of class 1 for such patients is closer to 0.5, then their risk of developing diabetes is higher, indicating a less certain classification.

The diabetes score was computed by multiplying the probability estimate for class 1 by 100, representing the percentage likelihood of a patient developing diabetes. Despite being classified as non-diabetic, the patients had high probabilities for class 1 prediction, indicating a high risk of developing diabetes.

To evaluate the accuracy of the probability estimates, data from patients at the start of the study were compared with their data after a 2-year period. Two participants in the control group were not informed of their probability and diabetes score, and they continued with their unhealthy habits of poor diet and lack of physical activity. On the other hand, two participants in the experimental group were informed of their probability and diabetes score, and they changed their habits by adopting healthier diets and exercising regularly. The hypothesis was that patients who developed diabetes would have an increase in the probability of class 1 or diabetes prediction for their 2-year data, while patients who did not develop diabetes would have a decrease in the probability of class 1 or diabetes prediction for their 2-year data.

### 2.4. Statistical Analysis

Various statistical methods were applied to analyze and assess the performance of the machine learning models, as described above. Pearson’s correlation coefficient was utilized to measure the strength and nature of the linear association between variables. The calculation of Pearson’s correlation coefficient was facilitated by the implementation of appropriate functions from the NumPy library in Python. Furthermore, K-fold cross-validation, accuracy, precision, recall, and f1-score were employed as statistical metrics to evaluate the predictive capabilities of the models. The Scikit-learn library in Python was leveraged to compute these metrics. By utilizing the functionalities provided by the Scikit-learn and NumPy libraries, the study successfully executed these statistical methods for comprehensive model evaluation and analysis.

## 3. Results

### 3.1. Model Selection

#### 3.1.1. Models Performance

Table 4 and Figure 3 report the mean and standard deviation of performance metrics for various machine learning algorithms on the analyzed dataset using cross-validation. The mean value reflects the arithmetic average of the algorithm’s performance over all the data folds, while the standard deviation indicates the degree of dispersion of the performance measure around the mean value across different folds. These statistical indicators provide a quantitative assessment of the performance of the algorithms and the degree of variability in their performance estimates. The mean values of the performance metrics of the machine learning algorithms were calculated and analyzed. It was observed that the Random Forest Classifier and Gradient Boosting Classifier had the highest average score of 0.998, followed by the Ada Boost Classifier and the Decision Tree Classifier algorithms with mean scores of 0.995 and 0.989, respectively. In contrast, the SVC and K Neighbors Classifier algorithms had the lowest mean performance scores of 0.787 and 0.842, respectively.

The standard deviation values provide insight into the consistency of algorithm performance across different folds. Lower standard deviation values signify greater consistency, whereas higher values indicate more variability in performance. The Random Forest Classifier and Gradient Boosting Classifier algorithms demonstrated the most stable performance across all folds, as they had the lowest standard deviation value of 0.005. On the other hand, the Logistic Regression algorithm had the highest standard deviations of 0.050, indicating greater variability in their performance across different folds.

The performance measure values of all the classification algorithms are shown in Figure 4. In Figure 4, we can see that the accuracy of all classification methods is above 70%. The results showed that the Random Forest Classifier had the highest accuracy of 100%, while conversely, the SVC had the lowest accuracy of 79%.

#### 3.1.2. Hyperparameter Tuning

Based on the training and testing results of the models, Random Forest Classifier achieved the highest accuracy and was the best-fitting algorithm for our dataset. Therefore, it was chosen for hyperparameter tuning.

The model was trained using the following hyperparameters: a maximum depth of 3, 5, or 7, a minimum number of samples required to be at a leaf node of 1, 2, or 4, a minimum number of samples required to split an internal node of 2, 5, or 10, and a number of trees in the forest of 50, 100, or 200.

Based on the results of cross-validation, the random forest model appears to be the most effective when utilizing the hyperparameters of {‘max_depth’: 7, ‘min_samples_leaf’: 1, ‘min_samples_split’: 2, ‘n_estimators’: 50}. The mean accuracy score of 0.9984 suggests that the model is capable of accurately predicting the target variable for a significant proportion of the samples within the dataset. Furthermore, the relatively low standard deviation of 0.0032 indicates that the model is consistently accurate across the various folds of the cross-validation process.

Overall, the results of the cross-validation process suggest that the random forest model with the selected hyperparameters is a strong candidate for accurately predicting the target variable for new, unseen data.

### 3.2. Machine Learning Model’s Performance in Assisting Diabetes Diagnosis

The system was utilized in conjunction with medical professionals to diagnose 67 new patients. The physicians ascertained that 34 of the patients were diagnosed with diabetes, whereas 33 patients were not diagnosed with diabetes, and these results were utilized as the ground truth outcome for comparison purposes (Figure 5).

Among the 34 diabetes cases, the model successfully 30 cases, resulting in a true positive rate of 88.24%, which means that the model correctly identified 88.24% of patients with diabetes. The positive predictive value is 100%, which means that out of the 30 patients that the model identified as having diabetes, all of them actually had the disease.

However, four patients with diabetes were misclassified as negative, which is referred to as false negatives. Therefore, the negative predictive value was 89.2%, which indicates that when the model predicted no diabetes, it was correct 89.2% of the time.

On the other hand, the model successfully detected all 33 other cases of non-diabetes, resulting in a 100% specificity. However, the negative predictive value (NPV) of the model is 89.2%, meaning that there were four false-negative cases out of 37 patients who did not have diabetes.

In other words, as shown in Table 5,the precision of class 0 is 0.89, meaning that out of all the patients predicted to not have diabetes, 89% of them truly do not have the disease. The recall of class 0 is 1.00, which means that the model correctly identified all patients who did not have diabetes. In contrast, the precision of class 1 is 1.00, meaning that out of all the patients predicted to have diabetes, all of them truly do have the disease. However, the recall of class 1 is 0.88, indicating that the model missed identifying 12% of patients who actually had diabetes.

The AUC value of 0.94 in the ROC curve suggests that the model has a relatively high true positive rate (sensitivity) and a relatively low false positive rate (1-specificity) across various threshold values, indicating that it has a good ability to distinguish between positive and negative samples (Figure 6). The model’s overall accuracy of 0.94, and the average precision, recall, and f1-score of 0.95, 0.94, and 0.94, respectively, also support this interpretation.

### 3.3. Quantification of Diabetes Risk and Application

The study found that the traditional diagnostic approach to diabetes, which relies on a binary diagnosis, can hinder the management and prevention of diabetes-related complications, especially for patients with limited health literacy or numeracy skills, and highlights the potential of machine learning algorithms to provide more precise and quantitative approaches to diabetes diagnosis and risk assessment.

In specifics, Table 6 provided in the study shows the before and after data for Patient 1 and Patient 2, who were part of the control group. The patients were initially classified as non-diabetic with a class 1 probability of 0.47 and 0.44, respectively, and diabetes scores of 47 and 44, respectively. This probability indicates a high risk of developing diabetes. The patients relied on their physician’s diagnosis, which is typically binary and does not provide a comprehensive representation of the patient’s condition. After two years, the patients were diagnosed with diabetes, and the AI predictions showed a high probability of class 1 at 0.92 and 0.97.

In contrast, Patient 3 and Patient 4, who were part of the experimental group, were initially classified as non-diabetic with a class 1 probability of 0.44 and 0.33, respectively (Table 7). They were informed of their diabetes score and made necessary lifestyle changes to prevent the progression of their illness. After two years, their class 1 probability decreased to 0.33 and 0.28, and their diabetes scores reduced to 33 and 28, respectively. These results suggest that the model successfully predicted the probability of patients developing diabetes, as shown by the changes in class 1 probabilities and diabetes scores, which are indicators of high risk for developing diabetes.

## 4. Discussion

### 4.1. Machine Learning Model’s Performance in Assisting Diabetes Diagnosis

According to the results, it was observed that the Random Forest Classifier was the best-fitting model for our dataset, as it had achieved the highest cross-validation average score of 0.998, with the lowest standard deviation value of 0.005 and the highest accuracy of 100%. To ensure the model’s generalizability, we evaluated its performance on a separate test set.

The obtained accuracy of 94% in the current study demonstrates the model’s high level of accuracy in identifying patients with diabetes. This result is particularly promising, as it is comparable to the accuracy rates reported in previous research studies. For instance, a study conducted by Xu et al. utilized a random forest model on a dataset from the School of Medicine, University of Virginia. This dataset consisted of 403 testers and 19 features related to factors such as age, sex, cholesterol, hemoglobin, waist, and hip. The study reported an accuracy rate of 85.00% for the random forest model, which outperformed other models, such as the ID3 algorithm (78.57%), Naive Bayes algorithm (79.89%), and AdaBoost algorithm (84.19%) [37]. Similarly, in a study by Benbelkacem and Atmani, the random forest algorithm was applied to the Pima Indians Diabetes dataset, resulting in a low error rate of 0.21 and an accuracy of 79% [38]. Furthermore, Kumari and Chitra employed a Support Vector Machine (SVM) algorithm to develop a predictive model for diagnosing diabetes using the Pima Indian diabetic database. The SVM model achieved an overall accuracy of 78% in correctly identifying patients with and without diabetes. The model demonstrated a sensitivity of 80% and specificity of 76.5% [39]. Comparing these studies, the current research showcases a higher accuracy rate than the mentioned models, indicating its effectiveness in accurately identifying patients with diabetes. The findings highlight the significant progress made in the field of machine learning algorithms for diabetes diagnosis and emphasize the potential of the developed model in improving patient outcomes and facilitating personalized healthcare.

Despite achieving high overall accuracy, the Random Forest Classifier demonstrated a higher proficiency in predicting class 0 (non-diabetic) cases compared to class 1 (diabetic) cases. The model successfully identified all patients without diabetes, indicating a low rate of false positives. However, it had a 12% false negative rate, meaning that a portion of patients with diabetes were incorrectly classified as non-diabetic. This discrepancy suggests that the model faced challenges in accurately predicting cases of diabetes, resulting in a higher number of missed diagnoses among patients with the disease. The observed discrepancy in the model’s performance can be attributed to the inherent complexity and variability of diabetes itself. Diabetes is a multifaceted condition influenced by a combination of genetic predisposition, lifestyle choices, and environmental factors. The intricate nature of diabetes makes it challenging to capture all cases accurately, particularly within the context of predictive modeling. Furthermore, imbalances in the distribution of class labels or variations in the representation of features associated with diabetes can impact the model’s learning process and subsequent classification performance. To better understand and interpret these results, it is essential to consider the complexities of diabetes and the inherent challenges associated with predictive modeling in this domain. Continued research efforts can focus on refining the model through various approaches, such as feature selection, algorithm optimization, and dataset augmentation. By addressing these aspects, it is possible to enhance the model’s performance and accuracy in detecting all cases of diabetes.

However, even though the model may have some limitations in detecting all cases of diabetes, it is important to note that it is not uncommon for screening tests to have some false negatives. Indeed, the ROC curve and the AUC of 0.94 still indicated a good ability to distinguish between positive and negative samples, with a relatively high true positive rate and a relatively low false positive rate across various threshold values. This is an indication of the model’s ability to correctly identify patients with diabetes and minimize the occurrence of false positives. The high specificity rate achieved by the model in this study indicates that it is highly accurate in identifying patients who do not have diabetes. This means that the model can be a useful tool for ruling out the presence of diabetes in patients who may have been flagged as at risk through other screening methods. This can lead to significant cost savings for patients and healthcare systems, as unnecessary diagnostic tests and treatments can be avoided. By focusing resources on patients who are more likely to have diabetes, healthcare providers can optimize the allocation of resources, reduce healthcare expenditures, and alleviate the financial burden experienced by patients. Especially in resource-limited settings where the healthcare infrastructure is not well-developed, and the number of healthcare professionals is inadequate, the use of the model can aid in patient screening and alleviate the burden for the patients, as well as on the local healthcare system. Additionally, the model’s high specificity rate contributes to improving patient care and overall health outcomes. Patients who receive a negative classification for diabetes can be reassured that they are unlikely to have the disease, providing them with peace of mind and alleviating unnecessary anxiety or worry. This can foster a better doctor–patient relationship and enhance patient satisfaction. Furthermore, the model’s specificity can be leveraged to guide further preventive measures and interventions. Patients identified as non-diabetic by the model can be advised on lifestyle modifications and preventive strategies to maintain their current health status and minimize their future risk of developing diabetes. This personalized approach empowers patients to take control of their health and make informed decisions about their well-being.

Overall, while the model’s performance may be slightly better for identifying patients without diabetes, the model performed well in accurately identifying patients with diabetes and non-diabetes and can still assist clinicians in diabetes diagnosis and patient care by providing additional information and insights to aid in decision-making. Clinicians can use the model’s predictions as a reference, along with other clinical and diagnostic information, to make informed decisions about patient care and treatment plans.

### 4.2. Quantification of Diabetes Risk and Application in Assisting Diabetes Diagnosis

The present study investigated the ability of an AI model to predict the risk of developing diabetes in patients. The study included four patients, with Patients 1 and 2 in the control group and Patients 3 and 4 in the experimental group. Patients 1 and 2 were initially classified as non-diabetic but had high-risk scores for developing diabetes. They relied on a binary physician diagnosis and were diagnosed with diabetes after two years. In contrast, patients 3 and 4 had lower risk scores and were provided with information on their diabetes scores to make lifestyle changes. After two years, their risk scores decreased. Overall, the model successfully predicted the risk of developing diabetes, as indicated by the changes in risk scores and class 1 probabilities, confirming the hypothesis.

Diabetes detection and diagnosis have traditionally relied on a doctor’s experience, knowledge, and subjective interpretation of the symptoms and test results. However, the diagnostic approach to diabetes utilized by physicians is typically binary, providing a diagnosis of either the presence or absence of the disease. At best, the physician may provide an assessment of disease severity through the identification of pre-diabetes, which also relies on a binary diagnosis of either present or absent. However, this approach does not provide a comprehensive representation of the patient’s condition, as it fails to capture the continuum of disease severity that exists between the binary diagnoses. Diabetes is a complex and multifaceted disease, and by solely relying on a binary diagnosis, clinicians may overlook important nuances and fail to address the specific needs of patients. Additionally, even though the patient’s level of glycemic control and risk of diabetes and complications can be identified through test results, interpreting the results of these tests and understanding how to manage and improve glycemic control can be challenging for some patients, especially those with limited health literacy or numeracy skills. As a result, their glycemic control may be suboptimal, leading to increased risks of complications associated with diabetes. These limitations in the traditional diagnostic approach can impede the effective management and prevention of diabetes-related complications. Without a comprehensive understanding of the patient’s condition and tailored interventions, healthcare providers may not be able to provide the most appropriate treatments and preventive measures. This can result in suboptimal disease management, increased risks of complications, and reduced overall quality of care.

In contrast, the results indicated that the ML algorithm provided a more precise and quantitative approach to diabetes diagnosis and risk assessment. The study notes that class 1 probabilities can be considered indicators of high risk for developing diabetes due to their association with the likelihood of an individual having the condition. Class 1 refers to the positive or “diabetic” class in a binary classification problem, meaning that a high class 1 probability indicates a higher likelihood of the individual belonging to the diabetic class. Therefore, the model was successful in quantifying the risk of diabetes for patients. The ability to communicate the risk of diabetes in quantitative terms offers significant advantages for healthcare professionals in their clinical practice. By utilizing numerical risk assessments, healthcare professionals can effectively convey the magnitude and implications of diabetes to patients, facilitating better understanding and engagement. This quantitative approach supports shared decision-making, enabling patients to actively participate in treatment discussions based on their precise risk levels. Healthcare professionals can tailor interventions and treatment plans according to individual patient risks, prioritizing resources and setting realistic goals. For instance, patients at high risk may receive more intensive monitoring, counseling, or interventions to address modifiable risk factors. On the other hand, patients at lower risk may require fewer intensive interventions, focusing more on lifestyle modifications and preventive measures. Quantitative risk assessments also aid in monitoring disease progression over time, allowing for proactive management and timely adjustments to the management plan.

Overall, the results indicated that the model’s ability was successful in predicting and quantifying the probability of a patient developing diabetes through the class 1 probability and the resulting diabetes score. The model is not only able to identify high-risk patients but can also track changes in their risk status over time. The ability to quantify diabetes risk using a predictive model can aid in early detection, prevention, and management of diabetes, ultimately improving patient outcomes. By using this information, healthcare providers help patients better understand their condition, as well as make informed decisions about patient care and adjust their treatment plans accordingly and potentially prevent the onset of diabetes or improve the management of existing diabetes.

### 4.3. Limitation of the Study

The study has several limitations that should be acknowledged. One potential limitation of the study is the potential presence of unaccounted factors that may influence the outcomes of the machine learning model in detecting type 2 diabetes. Factors such as genetic predispositions, lifestyle behaviors, or comorbidities could contribute to the development of diabetes and impact the accuracy of the model’s predictions. Additionally, the study faced challenges with missing data due to dropouts and resource constraints, resulting in the deletion of instances and reducing the available dataset size. Moreover, the limited number of diabetes cases in the dataset may have affected the model’s ability to accurately detect positive cases, potentially leading to overfitting. To address this, over-sampling techniques were employed to balance the class distribution. Despite these limitations, the study provides valuable insights into the application of machine learning for diabetes detection, but further research is warranted to address these limitations and enhance the generalizability and reliability of the findings.

## 5. Conclusions

The investigation indicates that the implementation of machine learning algorithms can offer more refined and quantifiable strategies for diabetes diagnosis and risk evaluation. Specifically, the Random Forest Classifier model exhibits considerable precision in identifying diabetes and, therefore, exhibits potential as a valuable instrument for clinicians in diagnosing diabetes. Additionally, the study’s key finding highlights the algorithm’s accurate prediction of the likelihood of patients developing diabetes or diabetes score, demonstrated by the class 1 (diabetes) probabilities. Overall, this study emphasizes the importance of harnessing machine learning in healthcare and diabetes diagnosis to enhance early detection, implement targeted interventions, and empower patients to actively manage their health. Continued research and integration of these technologies have the potential to transform healthcare delivery, improve patient outcomes, and alleviate the global burden of diabetes.

## Figures and Tables

**Figure 1 diagnostics-13-02087-f001:**
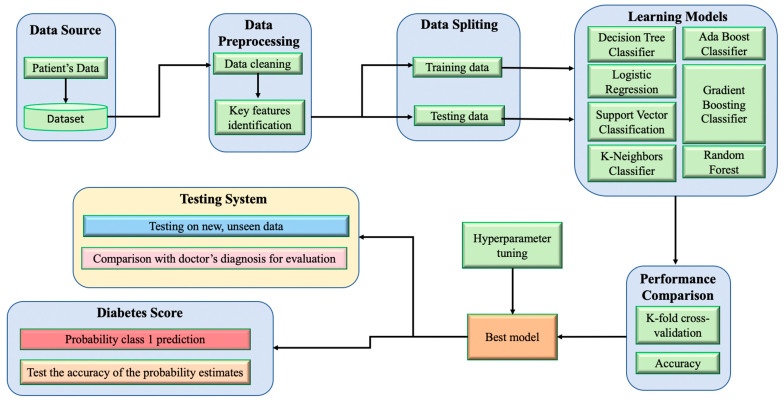
The framework of machine learning.

**Figure 2 diagnostics-13-02087-f002:**
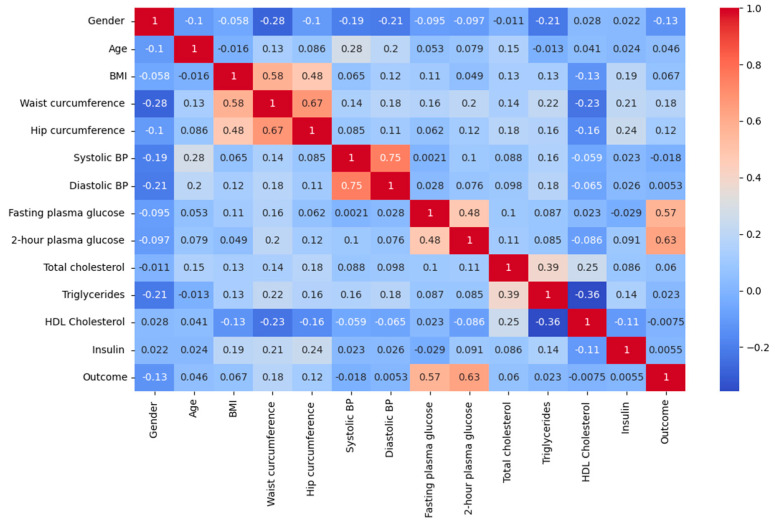
Correlation heatmap.

**Figure 3 diagnostics-13-02087-f003:**
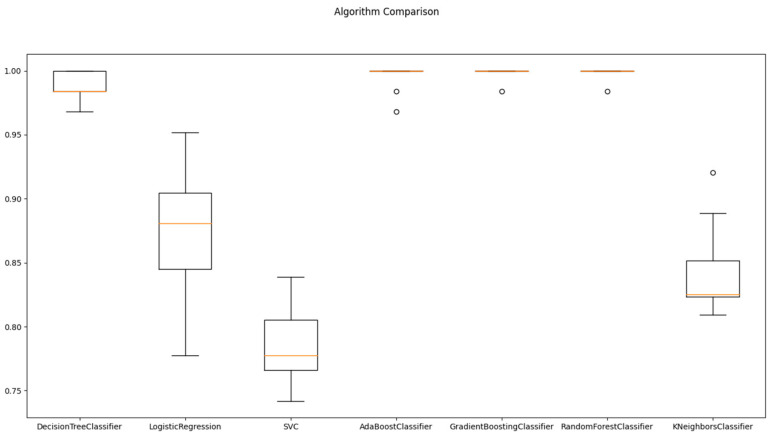
Algorithms comparison for Train/Test splitting method.

**Figure 4 diagnostics-13-02087-f004:**
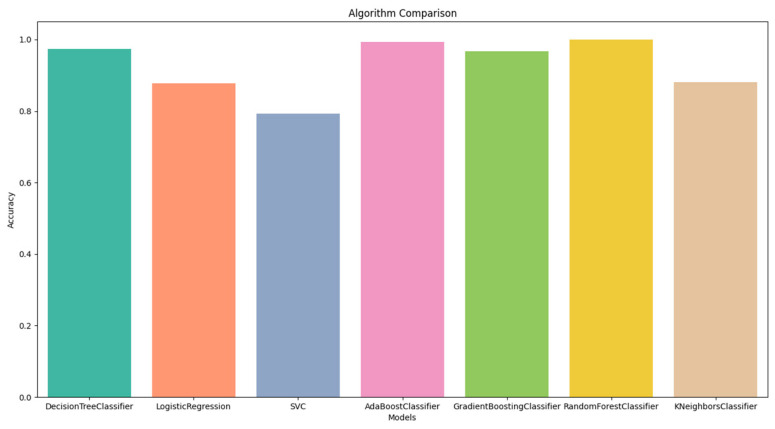
Algorithms comparison using Train/Test splitting method.

**Figure 5 diagnostics-13-02087-f005:**
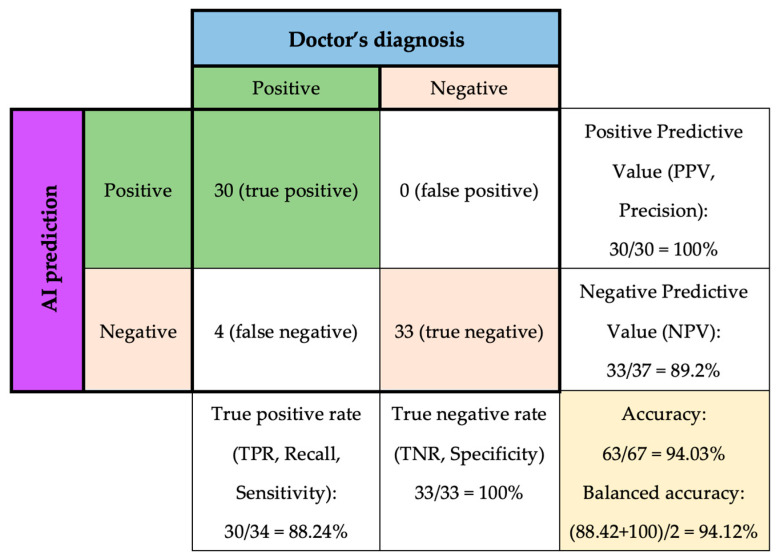
Tuned Random Forest Classifier performance on predicting new data.

**Figure 6 diagnostics-13-02087-f006:**
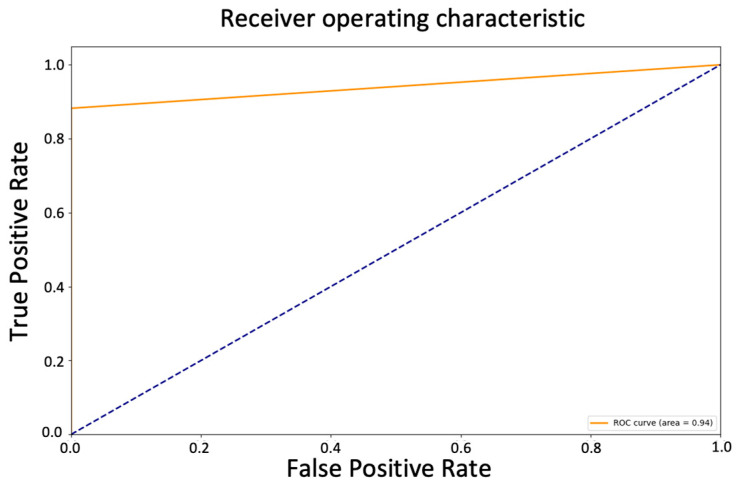
ROC curve of tuned Random Forest Classifier predicting new data.

**Table 1 diagnostics-13-02087-t001:** The attributes of the dataset.

Attributes	Range	Description
Gender	1–2	Male or female; 1 = male, 2 = female
Age	30–87.97	Age in years
BMI	13.63–39.45	Body mass index (BMI) = (weight in kg/(height in m)^2^)
Waist circumference	54–112	Anthropometric measurement of the circumference of the waist at the point halfway between the lowest rib and the top of the hip bone (cm)
Hip circumference	52–120	Anthropometric measurement of the maximum girth at the level of the greater trochanters (cm)
Blood Pressure (BP)	84–202 (systolic) 42–123 (diastolic)	Diastolic and systolic blood pressure (mmHg)
Fasting plasma glucose	2.3–13.0	Blood sugar levels after fasting or not eating anything for at least 8 h (mmol/L)
Plasma glucose after 2 h	2.6–24.4	Plasma glucose concentration in a 2-h oral glucose tolerance test (mmol/L)
Total cholesterol	0.3–15.4	Total amount of cholesterol present in the blood (mmol/L)
Triglycerides	0.35–15.5	Amount fat molecule found in the blood (mmol/L)
HDL Cholesterol	0.48–2.5	Amount of high-density lipoprotein (HDL) present in the blood (mmol/L)
Insulin	9.58–529.6	Total amount of insulin secreted by the pancreas in response to increased levels of glucose in the blood (pmol/L)
Outcome	0–1	Class variable, diagnoses classes: 0 = no diabetes, 1 = diabetes

**Table 2 diagnostics-13-02087-t002:** The number of missing values.

Attributes	No. of Missing Values
Gender	0
Age	396
BMI	364
Waist circumference	233
Hip circumference	245
Systolic BP	376
Diastolic BP	376
Fasting plasma glucose	328
2-h plasma glucose	448
Total cholesterol	1294
Triglycerides	1295
HDL Cholesterol	1296
Insulin	1295
Outcome	0

**Table 3 diagnostics-13-02087-t003:** The correlation between input and output attributes.

Attributes	Correlation Coefficient
Gender	−0.13
Age	0.046
BMI	0.067
Waist circumference	0.18
Hip circumference	0.12
Systolic BP	−0.018
Diastolic BP	0.0053
Fasting plasma glucose	0.57
2-h plasma glucose	0.63
Total cholesterol	0.06
Triglycerides	0.023
HDL Cholesterol	−0.0075
Insulin	−0.0055
Outcome	1.000000

**Table 4 diagnostics-13-02087-t004:** The performance measure of all classification methods for K-fold cross-validation.

Algorithms	Mean	Standard Deviation
Decision Tree Classifier	0.989	0.010
Logistic Regression	0.876	0.050
SVC	0.787	0.029
Ada Boost Classifier	0.995	0.010
Gradient Boosting Classifier	0.998	0.005
Random Forest Classifier	0.998	0.005
K Neighbors Classifier	0.842	0.034

**Table 5 diagnostics-13-02087-t005:** Confusion matrix of tuned Random Forest Classifier predicting new data.

	Precision	Recall	F1-Score	Support
0	0.89	1.00	0.94	33
1	1.00	0.88	0.94	34
accuracy			0.94	67
macro avg	0.95	0.94	0.94	67
weighted avg	0.95	0.94	0.94	67

**Table 6 diagnostics-13-02087-t006:** Prediction and diabetes probabilities.

		Patient 1		Patient 2	
		Before	After	Before	After
Gender		1	1	2	2
Age		60	62	58	60
BMI		24.46	25.32	24.61	24.31
Waist circumference (cm)		90	88	91	96
Hip circumference (cm)		96	94	99	104
Systolic BP (mmHg)		160	155	116	109
Diastolic BP (mmHg)		94	107	75	73
Fasting plasma glucose (mmol/L)		6.6	6.3	6.5	6.6
2-h plasma glucose (mmol/L)		11	16.7	10.4	11.7
Total cholesterol (mmol/L)		6.1	6.5	4.8	5.3
Triglycerides (mmol/L)		0.93	1.2	0.78	0.9
HDL Cholesterol (mmol/L)		1.17	1.15	1.17	1.07
Insulin (pmol/L)		21.2	24.1	12.6	15.4
Doctor’s diagnosis		0	1	0	1
AI prediction	prediction	0	1	0	1
	0 probability	0.53	0.08	0.56	0.03
	1 probability	0.47	0.92	0.44	0.97
	Diabetes score	47	92	44	97

**Table 7 diagnostics-13-02087-t007:** Prediction and diabetes probabilities.

		Patient 3		Patient 4	
		Before	After	Before	After
Gender		1	1	1	1
Age		59	62	65	67
BMI		18.47	18.47	22.21	22.77
Waist circumference (cm)		68	78	74	75
Hip circumference (cm)		80	88	80	82
Systolic BP (mmHg)		127	126	150	143
Diastolic BP (mmHg)		72	45	81	79
Fasting plasma glucose (mmol/L)		5.8	5.2	6.1	6.4
2-h plasma glucose (mmol/L)		8.6	6.3	10.9	7.6
Total cholesterol (mmol/L)		4.2	4.4	7.6	6.1
Triglycerides (mmol/L)		0.49	0.35	7.7	7.5
HDL Cholesterol (mmol/L)		1.2	1.5	1.13	1.0
Insulin (pmol/L)		4.1	3.9	18.5	7.6
Doctor’s diagnosis		0	0	0	0
AI prediction	prediction	0	0	0	0
	0 probability	0.56	0.67	0.67	0.72
	1 probability	0.44	0.33	0.33	0.28
	Diabetes score	44	33	44	28

## Data Availability

The data used to support the findings of this study are available from the corresponding author upon request.

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
