# Peer review of "The Utilization of Machine Learning Algorithms for Assisting Physicians in the Diagnosis of Diabetes"

_diagnostics, 2023, doi:10.3390/diagnostics13122087_

Round 1

Reviewer 1 Report

The authors present their study on using ML for risk assessment of type 2 diabetes. I do have several comments for the authors.

1.  For the title, machine learning is a sub-type of artificial intelligence. so I would suggest the authors to revise the title to " The utilization of machine learning algorithms for assisting physicians in the diagnosis of type 2 diabetes.

2.  The Introduction section is too lengthy.  The discussion on the AI applications in healthcare should be removed. The authors should focus on the identified problems they try to work on and the hypothesis of their project.

3.  The way how the datasets were split for training and testing dataset should be justified.

4.  Was any statistical analysis performed for this study? If so, please include a separate paragraph for information of the statistical methods and software used for the project.

5.  The limitations of this study should be included in the Discussion Section.

Author Response

Reviewer 1:  Comments and Suggestions for Authors

The authors present their study on using ML for risk assessment of type 2 diabetes. I do have several comments for the authors.

  1. For the title, machine learning is a sub-type of artificial intelligence. so I would suggest the authors to revise the title to " The utilization of machine learning algorithms for assisting physicians in the diagnosis of type 2 diabetes.

Answer:

Thank you very much for your suggestion. We have revised the title of the paper

  1. The Introduction section is too lengthy.  The discussion on the AI applications in healthcare should be removed. The authors should focus on the identified problems they try to work on and the hypothesis of their project.

Answer:

Thank you very much for your comments. We have revised the Introduction section

  1. The way how the datasets were split for training and testing dataset should be justified.

Answer:

Our study employed a 70-30 split for training and testing the machine learning model. This means that 70% of the available data was used as the training set to train the model, while the remaining 30% of the data was used as the test set to evaluate the model's performance. The 70-30 split is a common practice in machine learning to strike a balance between having enough data for the model to learn from (training set) and having a sufficient evaluation set to assess the model's performance (test set). By using the 70-30 split, we ensured that the model had enough data for effective training while still having a separate dataset for robust evaluation, allowing them to assess the model's performance accurately and make reliable conclusions about its capabilities in predicting diabetes. This was included in part 4 of section 2.1.2

  1. Was any statistical analysis performed for this study? If so, please include a separate paragraph for information of the statistical methods and software used for the project.

Answer:

Various statistical methods, including Pearson's correlation coefficient, K-fold cross-validation, accuracy, precision, recall, and f1-score, were applied to analyze and assess the performance of the machine learning models in our study. We have included a section on the statistical methods and software used for statistical analysis in our study.

  1. The limitations of this study should be included in the Discussion Section.

Answer:

            We have added a section about the limitations of our study in the Discussion. Our research includes some limitations such as potential presence of unaccounted factors that may influence the outcomes of the machine learning model in detecting type 2 diabetes; the imbalance between the diabetes and non-diabetes cases; and the limited size of data

Reviewer 2 Report

The authors Ngupen et. al investigates the use of machine learning algorithms to aid medical professionals in the detection and risk assessment of diabetes.. This work is meaningful and difficult. However, in the reviewer opinion the paper needs some revisions to be recommendable for publication.

1. In abstract, the innovationsof the model and solution optimization should be illustrated.

2. In introduction, the research background of battery storage and hydroelectric topics in recent years should be added and completed. The following papers can provide some reference for problems. DOI:10.3390/pr11010042. 10.1016/j.ijmecsci.2023.108376. 10.1016/j.energy.2023.127015.  10.3390/pr11020568

3. The introduction should be organized. The innovation and difficult points should be highlight.

4. In section 4, the discussion and analysis should be expanded. And the figures 5-8 should be normative.

5. The model verification should be performed and illustrated.

6. What is the limitation of the proposed method.

No

Author Response

Reviewer 2:   Comments and Suggestions for Authors

The authors Nguyen et. al investigates the use of machine learning algorithms to aid medical professionals in the detection and risk assessment of diabetes. This work is meaningful and difficult. However, in the reviewer opinion the paper needs some revisions to be recommendable for publication.

  1. In abstract, the innovations of the model and solution optimization should be illustrated.

Answer:

Thank you very much for your suggestion. We have revised the abstract

  1. In introduction, the research background of battery storage and hydroelectric topics in recent years should be added and completed. The following papers can provide some reference for problems. DOI:10.3390/pr11010042. 10.1016/j.ijmecsci.2023.108376. 10.1016/j.energy.2023.127015.  10.3390/pr11020568

Answer:

Thank you very much for your suggestion. We have added these papers to ours

  1. The introduction should be organized. The innovation and difficult points should be highlight.

Answer:

Thank you very much for your suggestion. We have revised the Introduction section

  1. In section 4, the discussion and analysis should be expanded. And the figures 5-8 should be normative.

Answer:

Thank you very much for your suggestion. We have revised the Discussion section

  1. The model verification should be performed and illustrated.

Answer:

Our model was verified using Metrics Evaluation and K-fold cross-validation. Also, after identifying the most suitable algorithm for the dataset, we further evaluated the effectiveness of the selected model by testing it on a new dataset comprising 67 patients that had not been previously seen. The prediction of the model were compared to doctors’ diagnosis. We included in part 6 of section 2.1.2., and also section 2.2 in our paper.

  1. What is the limitation of the proposed method.

Answer:

We have added a section about the limitations of our study in the Discussion. Our research includes some limitations such as potential presence of unaccounted factors that may influence the outcomes of the machine learning model in detecting type 2 diabetes; the imbalance between the diabetes and non-diabetes cases; and the limited size of data.

Round 2

Reviewer 1 Report

The authors have addressed all my questions and concerns.

Author Response

Thank you for your comments.